# Butyrate Lowers Cellular Cholesterol through HDAC Inhibition and Impaired SREBP-2 Signalling

**DOI:** 10.3390/ijms232415506

**Published:** 2022-12-07

**Authors:** Stephanie Bridgeman, Hon Chiu Woo, Philip Newsholme, Cyril Mamotte

**Affiliations:** Curtin Health Innovation Research Institute and Curtin Medical School, Curtin University, Bentley, WA 6102, Australia

**Keywords:** HDAC inhibitors, butyrate, cholesterol, LDL, statins, SREBP-2

## Abstract

In animal studies, HDAC inhibitors such as butyrate have been reported to reduce plasma cholesterol, while conferring protection from diabetes, but studies on the underlying mechanisms are lacking. This study compares the influence of butyrate and other HDAC inhibitors to that of statins on cholesterol metabolism in multiple cell lines, but primarily in HepG2 hepatic cells due to the importance of the liver in cholesterol metabolism. Sodium butyrate reduced HepG2 cholesterol content, as did sodium valproate and the potent HDAC inhibitor trichostatin A, suggesting HDAC inhibition as the exacting mechanism. In contrast to statins, which increase SREBP-2 regulated processes, HDAC inhibition downregulated SREBP-2 targets such as HMGCR and the LDL receptor. Moreover, in contrast to statin treatment, butyrate did not increase cholesterol uptake by HepG2 cells, consistent with its failure to increase LDL receptor expression. Sodium butyrate also reduced ABCA1 and SRB1 protein expression in HepG2 cells, but these effects were not consistent across all cell types. Overall, the underlying mechanism of cell cholesterol lowering by sodium butyrate and HDAC inhibition is consistent with impaired SREBP-2 signalling, and calls into question the possible use of butyrate for lowering of serum LDL cholesterol in humans.

## 1. Introduction

Cardiovascular disease (CVD) is the leading cause of mortality worldwide [1], with elevations in serum low density lipoprotein cholesterol (LDL-C) being a major risk factor. The statin class of drugs are widely used to lower LDL-C, with an estimated >30 million statins users in the United States alone [2]. By inhibiting hydroxymethylglutaryl-CoA reductase (HMGCR), the rate-limiting enzyme in the mevalonate pathway of cholesterol biosynthesis, statins decrease cellular cholesterol, resulting in upregulation of the LDL receptor (LDLR) and thereby increasing the uptake of LDL from the bloodstream and lowering plasma LDL-C. Treatment with statins has proven to be effective in reducing morbidity and mortality from CVD by ~25% [3]. However, there is concern over their widespread use in those at the lower end of the CVD risk spectrum [4], especially since statin use has been associated with numerous adverse effects including myopathy, rhabdomyolysis, liver damage and more recently type 2 diabetes [3]. Asymptomatic, transient elevations in aminotransferases are the most common liver complaint with statin use, although statins use has also been associated with severe drug-induced liver injury. In particular, simvastatin and atorvastatin have each been implicated in over 60 published case reports of drug-induced liver injury and have both been associated with fatal liver injury [5]. As a result, there is interest in exploring alternative cholesterol-lowering agents with lower toxicity and diabetogenic risk.

Butyrate is a short chain fatty acid (SCFA) produced by the bacterial fermentation of dietary fibre and is an established inhibitor of histone deacetylases (HDACs). HDACs are a group of enzymes catalysing the deacetylation of histones and thereby conferring a less open chromatin structure, less amenable to gene transcription. HDACs influence a variety of metabolic pathways, and deregulation of HDACs has been associated with metabolic disorders including type 2 diabetes and CVD [6]. Given butyrate is produced by so called “good” gut bacteria, there is interest in whether it may have a role in conferring a health benefit and influence cholesterol metabolism. In animal models of metabolic diseases, supplementation with sodium butyrate has been reported to reduce serum triglyceride and cholesterol [7,8]. In vitro, butyrate has been found to lower cholesterol synthesis in the Caco-2 colon cancer cell line, associated with a decrease in HMGCR activity [9]. The effect of butyrate on cellular cholesterol levels in liver cells, the major site of cholesterol biosynthesis; macrophages, where cholesterol accumulation contributes to the development of foam cells and atherosclerotic plaques; or insulin-secreting beta cells, where altered cholesterol may impact insulin secretion and thus diabetes risk [10], has not been explored.

This study explores the effects of HDAC inhibition on cholesterol metabolism in various cell types, including liver cells, macrophages, and insulin-secreting cells. Aspects examined include influences on cell cholesterol content, the sterol regulatory element-binding protein 2 (SREBP-2) signalling pathway and cholesterol import and export.

## 2. Results

### 2.1. Sodium Butyrate Is Non-Toxic to HepG2 Cells

We previously demonstrated that statins reduce the viability of HepG2 cells, as determined by the reduction in resazurin using the Alamar Blue^®^ assay, in a dose-dependent manner [11]. Conversely, sodium butyrate does not significantly reduce cell viability at any of the tested concentrations, up to and including 20 mM (Figure 1).

### 2.2. Sodium Butyrate Reduces Cellular Cholesterol

In HepG2 cells, 5 mM sodium butyrate decreased cellular cholesterol content to a similar degree as treatment with 10 µM statins (Figure 2a). After 24 h treatment in media supplemented with lipoprotein deficient serum (LPDS), a lipoprotein deficient form of foetal bovine serum (FBS), sodium butyrate treated cells showed a 31% (±15% SEM) decrease in cholesterol content, comparable to the cholesterol lowering effect of the hydrophilic statins rosuvastatin (44% ± 15% SEM) and pravastatin (31% ± 14% SEM), as well as atorvastatin (52% ± 4% SEM) and simvastatin (53% ± 8% SEM). The reduction in cellular cholesterol by butyrate was significant from 3 mM, and similar results were seen with sodium valproate, a therapeutic anticonvulsant drug which is also an HDAC inhibitor and a SCFA (Appendix A).

Butyrate also reduced the cellular cholesterol content in BRIN-BD11 β cells (Appendix A), a commonly used model of insulin secreting pancreatic β cells which are robust, and which display high metabolic activity. However, in THP-1 macrophages, which had significantly lower cholesterol content overall, neither statins nor butyrate affected cholesterol content after 24 h (Appendix A). In a separate series of experiments, HepG2 cells were treated with both atorvastatin and sodium butyrate in order to determine if there were any additive or synergistic effects on cholesterol lowering. After 24 h treatment in LPDS, cells treated with both 5 mM sodium butyrate and 10 µM atorvastatin had comparable cholesterol content to cells treated with atorvastatin alone (Figure 2b).

Similar results were observed with cells treated in media containing the high lipoprotein fraction of FBS (HLPS), rather than the lipoprotein deficient FBS (LPDS) supplemented media used in the other experiments, with the magnitude of cholesterol lowering being lower than in LPDS, most likely due to increased uptake of cholesterol from the media. In HLPS, cellular cholesterol was lowered by 20%, 28% and 33% by sodium butyrate, atorvastatin and a combination of atorvastatin and sodium butyrate, respectively (Figure 2c). There was no significance difference between the cholesterol lowering effect of the three treatments.

In order to determine if the cholesterol lowering was a result of HDAC inhibition or free fatty acid receptor (FFAR) activation, HepG2 cells were treated with trichostatin A (TSA), a potent HDAC inhibitor, and sodium acetate, a SCFA and FFAR agonist with no known effect on HDAC activity [12]. After 24 h treatment with TSA, cellular cholesterol content was decreased by 20%, while sodium acetate had no effect (Figure 2d). Furthermore, there was no difference in cholesterol content between cells treated with a combination of TSA and sodium acetate compared to those treated with TSA alone, suggesting HDAC inhibition as the mechanism by which sodium butyrate lowers cellular cholesterol.

### 2.3. Sodium Butyrate Does Not Alter Cellular Triglyceride Content

In contrast to the effect on cholesterol, the triglyceride (TG) content of HepG2 cells was not affected by exposure to sodium butyrate for 24 h (Figure 3).

### 2.4. Sodium Butyrate Affects Neither Cholesterol Uptake nor Export

The primary mechanism by which statins lower circulating LDL-C is thought to be increased LDLR mediated LDL-C uptake by hepatocytes to compensate for the reduced cellular cholesterol biosynthesis due to HMGCR inhibition [13]. Fluorescently tagged cholesterol in the form of NBD-cholesterol was used to examine the influence of atorvastatin and sodium butyrate on cholesterol uptake in HepG2 cells (Figure 4a). As expected, 24 h treatment with atorvastatin increased the uptake of NBD-cholesterol, as did the included positive control treatment U-18666A, which inhibits cholesterol synthesis and intracellular cholesterol trafficking [14]. However, sodium butyrate had no effect on NBD-cholesterol uptake.

NBD-cholesterol was also used to examine cholesterol efflux. In these experiments, HepG2 cells were incubated with NBD-cholesterol for 48 h prior to treatment and fluorescence of the media was measured to detect exported NBD-cholesterol. Neither atorvastatin nor sodium butyrate affected the levels of NBD-cholesterol in media after 24 h treatment, although there was no positive control treatment in these experiments (Figure 4b). Furthermore, treatment with atorvastatin or sodium butyrate for 24 h had no effect on the secretion in the media of apolipoprotein A-I (Apo-A1) or apolipoprotein B (Apo-B), these being the major protein components of high-density lipoprotein (HDL) and LDL, respectively (Figure 5).

### 2.5. Sodium Butyrate and Statins Have Opposite Effects on SREBP-2 Signalling

SREBP-2 acts to increase intracellular cholesterol content by promoting the transcription of *LDLR* and cholesterol biosynthesis genes and is thought to be responsible for the increased LDL-C uptake seen in statin use [15]. In brief, when cell cholesterol levels are low, SREBP-2 is cleaved, releasing the active subunit for entry into the nucleus and subsequent activation of gene transcription.

In HepG2 cells, 24 h treatment with sodium butyrate lowered LDLR protein levels (Figure 6a) whereas atorvastatin increased LDLR levels, while neither treatment significantly altered protein levels of another SREBP-2 regulated target, HMGCR, or of the cleaved SREBP-2 protein (68 KDa fragment) itself (Figure 6b,c).

In THP-1 macrophages, sodium butyrate significantly lowered levels of cleaved SREBP-2 (Figure 6f), whereas reductions in LDLR (Figure 6d) and HMGCR (Figure 6e) were marginal, being significant using Student’s *t* test (*p* = 0.0001 and 0.0091, respectively) but not ANOVA (*p* = 0.078 and 0.39, respectively). This is in clear contrast with the influence of atorvastatin which increased LDLR expression substantially.

As sodium butyrate lowered active SREBP-2 without lowering cellular cholesterol in THP-1 macrophages, we theorised that butyrate may reduce SREBP-2 signalling at an earlier time point in HepG2 cells, lowering cholesterol and resulting in a compensatory increase in SREBP-2 signalling back to baseline levels. We therefore investigated SREBP-2 and HMGCR protein levels in HepG2 cells after 8 h and 16 h treatments. These experiments also included 16 h TSA treatment to determine if any effects seen were likely due to HDAC inhibition. After 8 h of exposure, sodium butyrate reduced LDLR (*p* < 0.05) but not HMGCR (not shown). At 16 h, sodium butyrate and TSA significantly lowered HMGCR protein levels (Figure 7a). Sodium butyrate tended to decrease cleaved SREPB-2 at 16 h (*p* = 0.08 at 5 mM), but the effect was clearer for the more potent HDAC inhibitor, TSA (Figure 7b). Gene expression studies using reverse transcription and quantitative PCR (RTqPCR) on cells treated with 5 mM sodium butyrate for 16 h showed reduced HMGCR mRNA (*p* = 0.05) (Figure 7c), but no effect on sterol regulatory element-binding protein 2 (*SREBF2*) mRNA levels (Figure 7d). This indicates that sodium butyrate likely alters SREBP-2 protein activity without altering its gene expression.

### 2.6. The Effect of Sodium Butyrate and Statins on Proteins Involved in Reverse Cholesterol Transport Depends on Cell Type and Media Lipoproteins

Reverse cholesterol transport involves the efflux of cholesterol from cells to the liver via HDL particles. ATP-binding cassette transporter sub-family A member 1 (ABCA1) mediates the transfer of cholesterol to apolipoprotein apo-A1 to form HDL, while scavenger receptor class B type 1 (SRB1) can bind HDL and allows for bidirectional transfer of cholesterol between cells and HDL [16]. In HepG2 cells treated in LPDS supplemented media, sodium butyrate decreased protein levels of both ABCA1 and SRB1, while atorvastatin also decreased ABCA1 (Figure 8a,c). When HepG2 cells were treated in HLPS supplemented media, i.e., using the high lipoprotein fraction of FBS, atorvastatin increased ABCA1 and tended to decrease SRB1 (*p* = 0.07), while sodium butyrate decreased ABCA1 and SRB1, as in LPDS (Appendix A). In BRIN-BD11 cells in LPDS supplemented media, sodium butyrate increased, and atorvastatin tended to decrease ABCA1 levels (significant by Student’s *t* test [*p* = 0.03] but not ANOVA [*p* = 0.17) (Figure 8b). In THP-1 macrophages in LPDS supplemented media, sodium butyrate increased levels of SRB1 (Figure 8d), while ABCA1 was decreased by both sodium butyrate and atorvastatin (Appendix A). The decrease in ABCA1 protein levels by sodium butyrate in HepG2 cells in LPDS supplemented media was evident from 8 h and was also observed with TSA, suggesting HDAC inhibition as the likely mechanism (Appendix A). The effect of sodium butyrate and statins on ABCA1 in different cell types and media conditions is summarised in Table 1.

### 2.7. Sodium Butyrate Alters the Expression of Multiple Cholesterol Related Genes

Sodium butyrate also had a significant influence on the mRNA levels of numerous cholesterol related genes additional to *LDLR* and *HMGCR* (Figure 9). In particular, the gene for ATP-binding cassette transporter sub-family G member 1 (ABCG1) was upregulated more than a hundred-fold in HepG2 cells, as was CAV1, the gene encoding caveolin-1, the key protein in caveolae lipid rafts. The genes for sterol regulatory element-binding protein 1 (*SREBF1*), SRB1 (*SCARB1*), and apo A-I (*APOA1*) were also upregulated, while those for ABCA1 *(ABCA1*) and cytochrome P450 family 7 sub family A member 1 (*CYP7A1*), encoding the rate-limiting enzyme in bile acid production, were downregulated. It should be noted that the robust upregulation of ABCG1 and CAV1 did not result in a significant increase in protein levels of ABCG1 or caveolin-1 (Appendix A), and SRB1 proteins levels were reduced with butyrate treatment despite the upregulation at the mRNA level (Figure 8c).

## 3. Discussion

By inhibiting HMGCR activity and thus reducing cell cholesterol content, particularly in the liver, statins cause cells to increase cholesterol uptake from the bloodstream, lowering plasma LDL-C. This is mediated by SREBP-2, which is activated, through cleavage, when cell cholesterol content decreases, resulting in increased expression of LDLR [15]. Our experiments confirmed atorvastatin treatment results in increased LDLR expression and cholesterol uptake in hepatic cells. Butyrate also lowered hepatic cell cholesterol, as did TSA, whereas the SCFA sodium acetate had no effect. This suggested the cholesterol lowering effect was due to HDAC inhibition, and thus that the cholesterol loweringmechanisms for statins and butyrate differ markedly. Furthermore, in contrast to atorvastatin, butyrate did not increase cholesterol uptake and tended to decrease SREBP-2 regulated events in hepatic cells, including protein expression for LDLR and HMGCR, albeit reaching significance at different time points of 8 h and 16 h, respectively. TSA, a more potent HDAC inhibitor, also decreased HMGCR and cleaved SREBP-2 protein levels at 16 h. Butyrate also reduced HMGCR and LDLR protein content in THP-1 macrophages, and this was linked to a trend to lower levels of activated or cleaved SREBP-2. The lack of cholesterol lowering in macrophages may be related to their lower cholesterol content under the conditions employed. While downregulation of HMGCR has been reported in butyrate treated intestinal cells, with studies finding reduced mRNA levels [17] and protein activity [9] in Caco-2 enterocytes, the discovery that this is mediated via HDAC inhibition is novel.

Our results also showed that butyrate lowered *HMGCR* but not *SREBF2* mRNA after 16 h treatment, i.e., that butyrate likely alters SREBP-2 activity but not the expression of its gene. Similar results were seen in a study of TSA treated neuronal cells, showing that HDAC inhibition by TSA reduced the expression of SREBP-2 regulated genes, including *LDLR* and *HMGCR*, and of active or cleaved SREBP-2 protein levels, but without significantly altering *SREBF2* gene expression [18]. Another possibility is that butyrate decreases *SREBF2* mRNA at an earlier timepoint than investigated in this study. Chittur, et al. [19] reported that TSA downregulated *SREBF2* mRNA expression in HepG2 cells with maximal repression occurring at 9 h treatment and levels normalising at 48 h. We cannot therefore rule out the possibility that the decrease in active SREBP-2 is due to decreased *SREBF2* gene expression, and time-course studies are warranted for future studies.

The SREBP-2 protein itself can be acetylated and this acetylation alters its localisation and activity [20]. In particular, the class III HDAC SIRT1 deacetylates SREBP-2, and SIRT1 inhibition increases levels of the active nuclear SREBP-2 protein and expression of SREBP-2 target genes [21]. Sodium butyrate does not inhibit class III HDACs [22], and it has not been investigated what effect class I and II HDACs, inhibited by butyrate, have on the acetylation of SREBPs, or whether inhibition of class I and II HDACs causes a compensatory increase in class III HDAC activity; both this, and direct evaluation of the binding of active SREBP-2 to the steroid response element of HMGCR and LDLR, for example by electrophoretic mobility shift assays, are aspects which could be examined in future studies [23].

While a decrease in SREBP-2 activity would explain the lowered cellular cholesterol, the fact that a lowering of cholesterol by butyrate in this manner cannot increase LDLR and increase the uptake of cholesterol by hepatic cells calls into question whether butyrate administration can lower serum LDL-C in a similar manner to statins (Figure 10).

On the other hand, one also needs to consider other possible indirect influences that orally administered butyrate may have on serum lipids, for example by stimulating the secretion of glucagon-like peptide 1 (GLP-1) [24,25]. The latter is an established enhancer of β cell function and insulin secretion [24,25], but there is also emerging evidence that GLP-1 agonists can lower serum LDL-C [26,27]. Therefore, while our in vitro studies give insights into the possible effect of butyrate on hepatic lipid metabolism in an in vitro setting, the question of whether butyrate can have beneficial effects on plasma lipids ultimately requires in vivo studies. To date however, findings from the few such studies conducted have been mixed. Of the in vivo rodent studies to date, some have found that butyrate supplementation reduced total plasma cholesterol [28,29,30], but other studies have found no influence on LDL-C [28,31]. A small placebo controlled human clinical trial on fifteen individuals per treatment group found that 45 days of butyrate supplementation increased GLP-1 secretion but did not significantly alter plasma cholesterol [32], and yet another small study, and without a placebo arm, found that butyrate increased LDL-C levels in subjects with the metabolic syndrome [33]. In contrast to humans, it is HDL rather than LDL which is the primary plasma cholesterol carrier in rodents [34], and such a difference could account for any discrepancies between rodent and human studies. A larger number of adequately powered trials are clearly needed to determine if butyrate supplementation alters plasma LDL-C in humans. The lowering of hepatic cholesterol seen with butyrate may be beneficial in and of itself, for example in NAFLD, where the accumulation of free cholesterol is thought to be a contributor to liver damage [35]. Indeed, animal models have shown sodium butyrate to be protective against diet-induced liver damage in mice [36,37].

The effect of sodium butyrate on proteins involved in HDL metabolism, namely ABCA1 and SRB1, varied according to cell type and the lipoprotein content of the media. ABCA1 mediates the efflux of cholesterol and phospholipids to apolipoprotein apo-A1 to form HDL particles [38]. ABCA1 expression is regulated by LXR, a nuclear receptor activated by oxysterols, cholesterol metabolites that occur in proportion to total cholesterol [39].

In the absence of ligands, LXR acts to repress target genes. Thus, ABCA1 is expressed proportionally to cellular cholesterol, providing a mechanism by which cholesterol lowering by both atorvastatin and sodium butyrate decrease ABCA1 levels in liver cells in low lipoprotein conditions, i.e., using LPDS. However, butyrate affected ABCA1 levels in disparate ways, lowering it in all tested conditions except in BRIN-BD11 cells. The fact that the decrease in ABCA1 also occurred in high lipoprotein conditions and in THP-1 macrophages, despite the lack of cholesterol lowering in these cells, suggests that butyrate can impact ABCA1 independent of cholesterol levels. As TSA also lowered ABCA1 protein levels, HDAC inhibition is likely the contributing mechanism. The downregulation of ABCA1 in macrophages could impair cholesterol efflux to HDL and thus increase cholesterol accumulation in these cells, while the upregulation of both ABCA1 and ABCG1 in insulin secreting cells suggests increased cholesterol efflux.

SRB1 acts as an HDL receptor and thus facilitates bidirectional movement of cholesterol between cells and HDL [40]. There is uncertainty over the transcriptional regulation of SRB1. Its promoter contains potential SREBP-1a, SREBP-2, [41] and LXR [42] binding sites. In macrophages, SRB1 levels have been reported to inversely correlate with cholesterol levels, while neither expression of constitutively active SREBPs nor knockout of LXR affected SRB1 expression [43]. It is also unclear how the increased macrophage SRB1 observed with butyrate may affect overall cholesterol levels; Ji and colleagues [16] found that SRB1 contributed to both efflux and influx of cholesterol between mouse bone marrow-derived macrophages and HDL, and as a result there was no overall difference in cholesterol contents between wild type and SRB1 knockout cells. Conversely, evidence suggests that in hepatic cells, SRB1 mediated cholesterol uptake exceeds cholesterol efflux; liver specific SRB1 overexpressing mice have lower plasma HDL and increased hepatic uptake of cholesterol esters from HDL [44], while liver specific SRB1 knockout mice have increased plasma HDL due to reduced hepatic uptake [45]. The downregulation of hepatic SRB1 and ABCA1 seen with sodium butyrate may therefore result in impaired RCT.

## 4. Materials and Methods

### 4.1. Preparation of Stock Solutions

Statins and TSA were sourced, prepared in dimethyl sulfoxide (DMSO) and stored at −20 °C or −80 °C for longer term storage, as previously described [24]. Sodium butyrate (Selleck Chemicals, Houston, TX, USA) and sodium acetate (Toronto Research Chemicals, Toronto, ON, Canada) were dissolved in ultra-pure distilled water to a concentration of 50 mM and stored at −20 °C.

### 4.2. Cell Culture and Treatment Conditions

HepG2 human hepatocellular carcinoma cells, BRIN-BD11 rat insulin-secreting β cells, and THP-1 human monocytes were maintained in tissue culture media supplemented with 10% FBS in 25 cm^2^ or 75 cm^2^ tissue culture flasks at 37 °C, in a humidified incubator equilibrated with 5% CO_2_ as previously described [11]. THP-1 cells were differentiated into macrophages by the addition of 50 nM phorbol 12-myristate 13-acetate (Sigma-Aldrich, St. Louis, MO, USA) to the culture media for 48 h, then allowed to recover for 24 h before treatments [11]. All cell treatments with statins or HDAC inhibitors were carried out in media supplemented with 10% LPDS, unless otherwise indicated, and conducted in a humidified incubator equilibrated with 5% CO_2_. Control cells were treated with media only for HDAC inhibitors or media containing DMSO as a vehicle control for statins.

### 4.3. Viability Assays

Following 22 h treatment of cells with sodium butyrate in 96 well plates, cell viability was assessed using alamarBlue^®^ (ThermoFisher, Waltham, MA, USA) as previously described [11]. Briefly, 10 µL of alamarBlue^®^ solution was added to the cells in treatment media, and following a 2 h incubation, fluorescence measurements were taken using excitation and emission wavelengths of 540 nm and 590 nm, respectively.

### 4.4. Lipid Quantification

After 24 h treatment in 96 well plates, media was removed, and cells were washed twice with PBS, extracted with hexane:isopropanol, and the cholesterol content quantified using the Amplex Red Cholesterol Assay Kit (ThermoFisher, Waltham, MA, USA) as previously described [24]. TG content was assessed using the High Sensitivity Triglyceride Fluorometric Assay Kit (Sigma-Aldrich) as per the manufacturer’s instructions. Samples were incubated with lipase for 20 min at 37 °C to hydrolyse TGs into glycerol and fatty acids. Samples were then incubated with the reaction master-mix containing the developer, probe and enzyme mix that reacts with the glycerol present. Following 30 min incubation at 37 °C, fluorescence was measured with an excitation wavelength of 535 nm and an emission wavelength of 587 nm. TG content of samples was determined using a standard curve based on a serial dilution of the TG standard provided with the kit.

### 4.5. Cholesterol Uptake and Export

Cholesterol uptake and export were determined using the Cholesterol Uptake Cell-Based Assay Kit (Cayman Chemical, Ann Arbor, MI, USA). For cholesterol uptake assays, cells were incubated in media containing 10% LPDS for 24 h prior to treatment containing 20 ug/mL of a fluorescently tagged cholesterol (NBD cholesterol, Cayman Chemical Ann Arbor, MI, USA). Following 24 h treatment, cells were washed in PBS, assay buffer was added, and fluorescence was measured with an excitation wavelength of 485 nm and an emission wavelength of 535 nm. The cholesterol transport inhibitor U-18666A was used as a positive control. For cholesterol export assays, cells in 96-well plates were incubated with media containing 10% LPDS and 20 ug/mL NBD cholesterol for 48 h prior to treatment in media containing 10% FBS. Following 24 h treatment, the media containing exported NBD cholesterol was transferred to fresh wells and fluorescence measured as for uptake assays.

### 4.6. Immunoblotting and Reverse Transcription Quantitative PCR

Immunoblotting and reverse transcription qPCR was conducted as we have previously described [24]. The primary antibodies used for immunoblotting are shown in Table 2, and the primer sequences and annealing temperatures used for qPCR of reverse transcribed mRNA are shown in Appendix A. Antibodies were sourced either from abcam (Cambridge, UK), or Santa Cruz Biotechnology (Dallas, TX, USA).

### 4.7. Statistical Analysis

Each experiment was conducted at least three times and independently. Statistical tests, including analysis of variance (ANOVA) or Student’s *t* test, as described in the text, were conducted using GraphPad Prism with *p* values < 0.05 considered significant.

## 5. Conclusions

This study describes the most comprehensive analysis of the effects of butyrate on cholesterol metabolism, in particular comparing and contrasting the effect to that of atorvastatin, a statin commonly used for lowering of plasma LDL-C. Previous animal studies have reported reduced serum cholesterol with butyrate supplementation, and it has been suggested that butyrate inhibits cholesterol biosynthesis in vitro. However, lowering of cellular cholesterol by butyrate has not previously been demonstrated. Furthermore, the results presented here suggest HDAC related influences, rather than FFAR activation, as the mechanism, with a time-dependent decrease in SREBP-2 signalling the most likely exacting mechanism. In contrast to statins, the cellular cholesterol lowering by butyrate does not increase uptake of cholesterol by hepatic cells, and thus calls into question whether butyrate could lower serum LDL-C in a similar manner.

The effects of butyrate on proteins involved in RCT were also in contrast to statins, where noted effects were directly related to cholesterol levels. The effects of butyrate on these proteins differed according to cell type, and as such it is difficult to draw conclusions on how this may affect HDL metabolism and RCT at the organismal level. Combined with the differences in cholesterol metabolism between humans and rodents, and the poor systemic availability of oral butyrate, there remain numerous doubts and questions over how butyrate may affect cholesterol metabolism in humans.

## Figures and Tables

**Figure 1 ijms-23-15506-f001:**
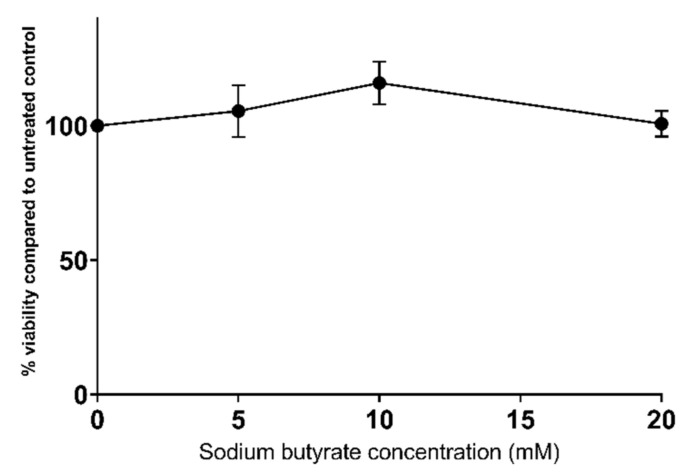
Sodium butyrate is not toxic to HepG2 cells when used at millimolar concentrations. Cells were exposed to butyrate for 24 h and viability assessed using the AlamarBlue^®^ assay. Results represent the mean from at least three independent experiments ± SEM.

**Figure 2 ijms-23-15506-f002:**
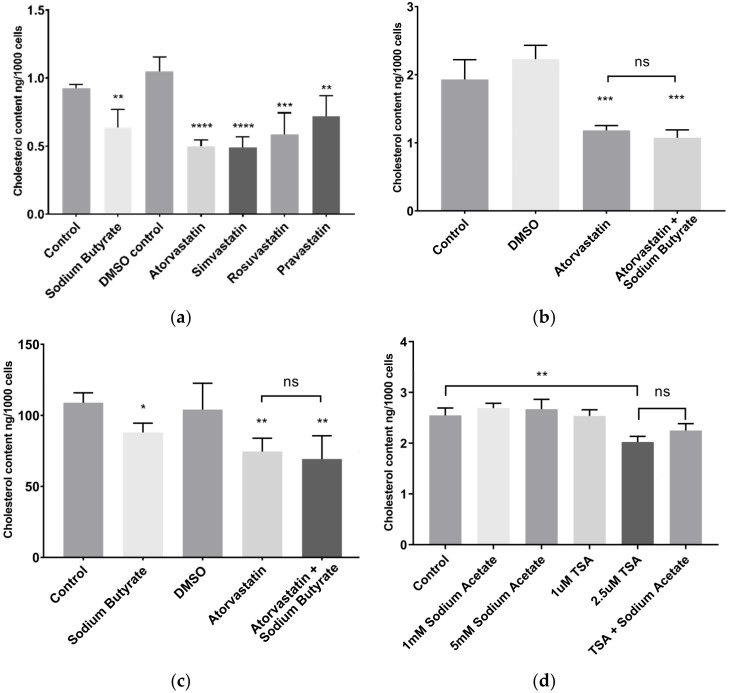
The cell cholesterol content of HepG2 cells is decreased by statins, and by the HDAC inhibitors, butyrate and TSA. The treatments were for 24 h and using: (**a**) 10 µM statins or 5 mM sodium butyrate in media supplemented with lipoprotein deficient serum (LPDS); (**b**) 10 µM atorvastatin with or without 5 mM sodium butyrate in media containing LPDS; (**c**) 10 µM atorvastatin, 5 mM sodium butyrate or in combination, in media supplemented with high lipoprotein serum (HLPS); (**d**) sodium acetate (SA) or trichostatin A (TSA) in media containing LPDS. The experimental control for butyrate treatment comprised of cells treated with media only (Control) and for statins, treatment with media containing dimethyl sulfoxide (DMSO). Results represent the mean from at least three independent experiments, with error bars representing SEMs. * *p* < 0.05, ** *p* < 0.01, *** *p* < 0.001, **** *p* < 0.0001, ns not significant.

**Figure 3 ijms-23-15506-f003:**
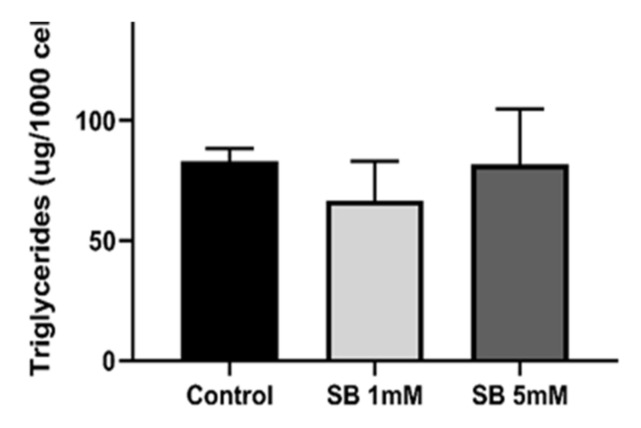
In contrast to its influence on cholesterol, butyrate has no influence on the triglyceride content of HepG2 cells. HepG2 cells were treated for 24 h with 1 mM or 5 mM sodium butyrate (SB) and cell triglyceride measured using the High Sensitivity Triglyceride Fluorometric Assay Kit (Sigma-Aldrich). Results represent the mean combined from at least three independent experiments, with error bars representing SEMs.

**Figure 4 ijms-23-15506-f004:**
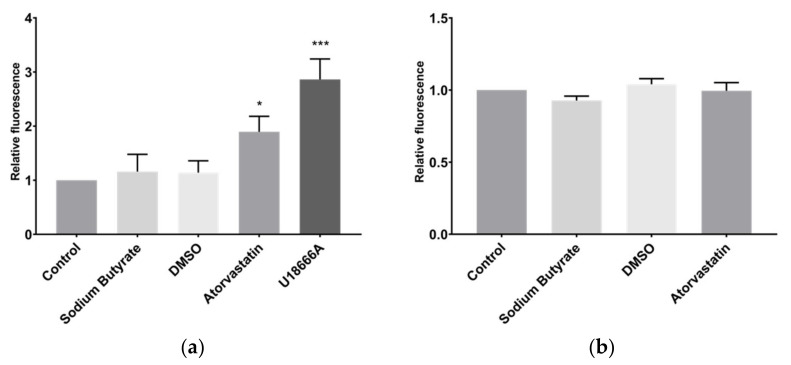
HepG2 cholesterol uptake is increased by atorvastatin and U18666A, but not by butyrate, and no treatment influenced cholesterol export. (**a**) Uptake of fluorescently tagged NBD-cholesterol by HepG2 cells; (**b**) Export of fluorescently tagged NBD-cholesterol from HepG2 cells. Cells were treated with 10 µM atorvastatin or 5 mM sodium butyrate for 24 h. Control cells were treated with media only for butyrate and media containing DMSO for atorvastatin. Results represent the mean from at least three independent experiments and error bars represent SEMs. * *p* < 0.05, *** *p* < 0.001.

**Figure 5 ijms-23-15506-f005:**
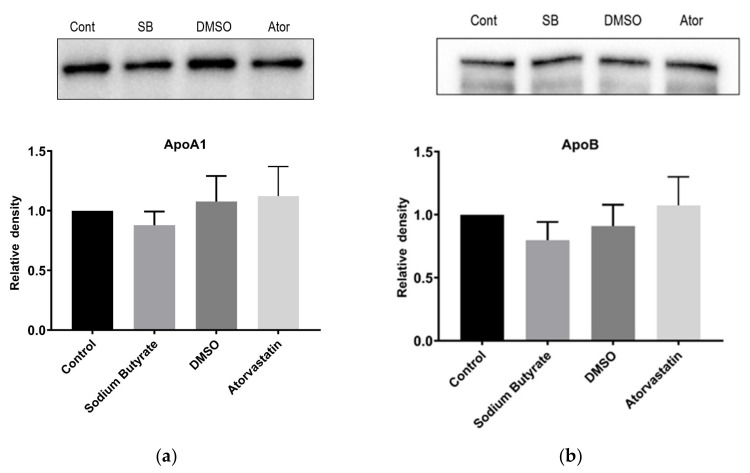
Butyrate and atorvastatin have no influence on apolipoprotein A-I and B secretion from HepG2 cells. Immunoblots of (**a**) Apo A-I and (**b**) Apo B in the media of cultured HepG2 cells following their treatment with 5 mM sodium butyrate and 10 µM atorvastatin for 24 h. Control cells were treated with media only for butyrate and media containing DMSO for atorvastatin. The graphs represent the mean density readings from three independent experiments, with error bars representing SEMs.

**Figure 6 ijms-23-15506-f006:**
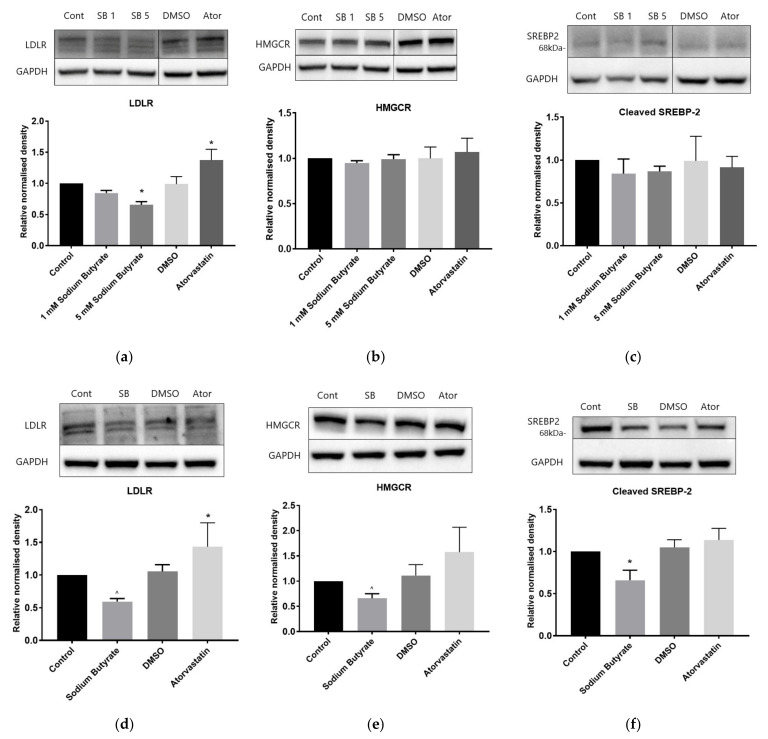
Atorvastatin and butyrate have differing effects on SREBP-2 signalling in HepG2 and THP-1 cells. In contrast to atorvastatin, 24 h treatment with sodium butyrate decreased rather than increased LDLR content in HepG2 and THP-1 cells. The 24 h treatment with butyrate also decreased HMGCR and cleaved SREBP-2 content in THP-1 cells, but not HepG2 cells. Immunoblots in HepG2 cells of (**a**) LDLR; (**b**) HMGCR; (**c**) cleaved SREBP-2; and in THP-1 cells of (**d**) LDLR (**e**) HMGCR; (**f**) cleaved SREBP-2. Cells were treated with 1 mM (SB1) or 5 mM (SB5) sodium butyrate (SB) or 10 µM atorvastatin for 24 h. Control cells were treated with media only for butyrate and media containing DMSO for atorvastatin. The graphs represent the mean density readings as normalised to GAPDH from three independent experiments, with error bars representing SEMs. * *p* < 0.05 using ANOVA, ^ *p* < 0.05 with two-tailed t test but not by ANOVA.

**Figure 7 ijms-23-15506-f007:**
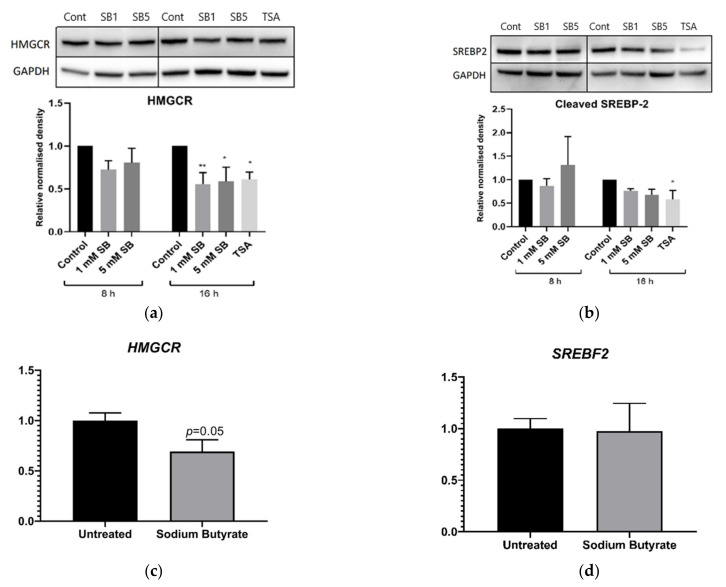
Influence of 8 and 16 h HDAC inhibition on cleaved SPREPB-2, HMGCR and *SREBF2* and *HMGCR* mRNA expression in HepG2 cells. In contrast to 24 h exposure, 16 h exposure to butyrate influenced HMGCR content in HepG2 cells, and the more potent HDAC inhibitor TSA decreased cleaved SREBP-2 content. Butyrate also diminished *HMGCR* gene expression but not expression of *SREBF2* which encodes SREBP-2. Immunoblots of (**a**) HMGCR and (**b**) cleaved SREBP-2. Cells were treated with 1 mM (SB1) and 5 mM (SB5) sodium butyrate (SB) for 8 and 16 h or 2.5 µM TSA for 16 h compared to the untreated control, normalised to GAPDH. Reverse-transcription qPCR for (**c**) *HMGCR* and (**d**) *SREBF2* mRNA expression. Cells were treated with 5 mM sodium butyrate for 16 h and compared to the untreated control, normalised to reference genes *GAPDH*, *YWHAZ* and *RPL13A*. Results represent the mean combined from at least three independent experiments, with error bars representing SEMs. * *p* < 0.05, ** *p* < 0.01.

**Figure 8 ijms-23-15506-f008:**
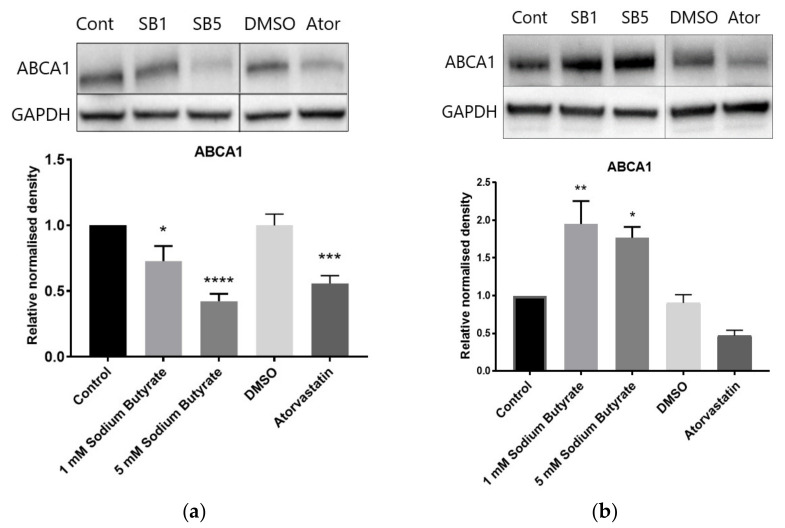
Influence of butyrate and atorvastatin on reverse cholesterol transport proteins in HepG2, BRIN-BD11 and THP-1 cells. Immunoblots of: (**a**) ABCA1 in HepG2 cells; (**b**) ABCA1 in BRIN-BD11 cells; (**c**) SBR1 in HepG2 cells; (**d**) SBR1 in THP-1 cells. Cells were treated with 1 mM (SB1) or 5 mM (SB5) sodium butyrate, or 10 µM atorvastatin for 24 h in media containing LPDS. Control cells were treated with media only for butyrate and media containing DMSO for atorvastatin The graphs represent the mean density readings as normalised to GAPDH from three independent experiments, with error bars representing SEMs. * *p* < 0.05, ** *p* < 0.01, *** *p* < 0.001, **** *p* < 0.0001 compared to control.

**Figure 9 ijms-23-15506-f009:**
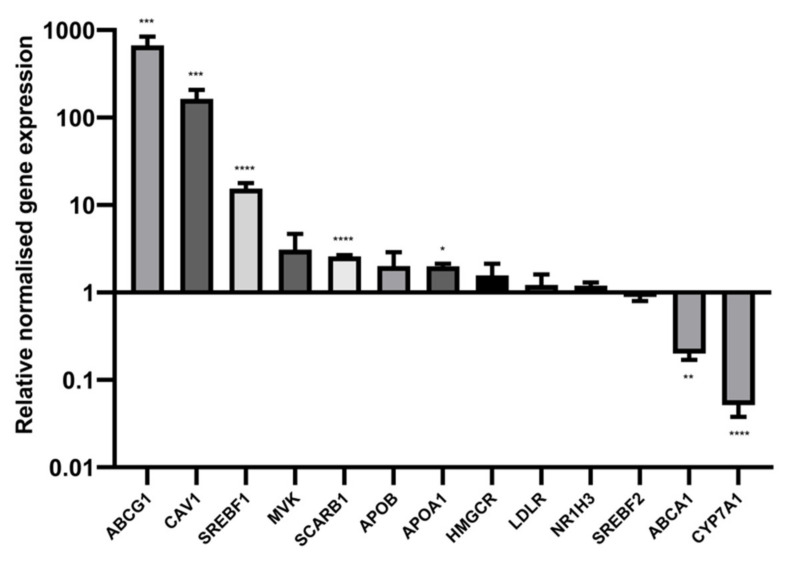
Influence of butyrate on cholesterol-related gene expression in HepG2 cells. Reverse-transcription qPCR results from HepG2 cells treated with 5 mM sodium butyrate for 24 h, normalised to reference genes *GAPDH*, *YWHAZ* and *RPL13A*. Results represent the mean combined from at least three independent experiments, with error bars representing SEMs. * *p* < 0.05 ** *p* < 0.01, *** *p* < 0.001, **** *p* < 0.0001.

**Figure 10 ijms-23-15506-f010:**
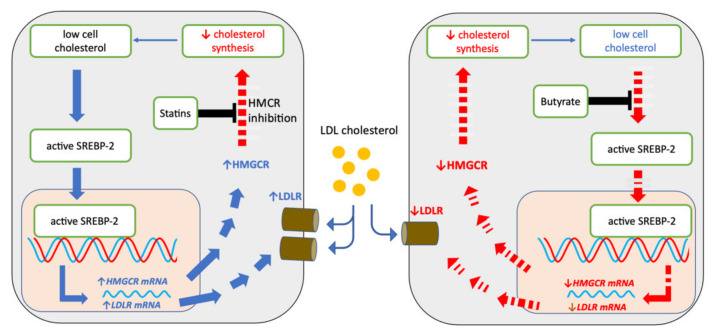
Influence of statins and that postulated for butyrate on cell cholesterol metabolism in HepG2 cells. Processes which are decreased with statin or butyrate treatment are shown in red. Statins inhibit HMGCR activity and thereby decreases de novo cholesterol synthesis, resulting in increased cleaved or active SREBP-2 which upregulates the expression of gene and protein expression of HMGCR and LDLR, the latter resulting in increased LDL cholesterol uptake. HMGCR protein is upregulated, but its activity remains inhibited by statins. Butyrate treatment is associated with decreased cleaved SREBP-2 activity resulting in decreased expression of HMGCR and LDLR gene and protein expression, resulting in decreased de novo synthesis and decreased LDL cholesterol uptake.

**Table 1 ijms-23-15506-t001:** Influence of butyrate and statins on ABCA1 protein levels in different cell types.

Treatment	HepG2 (LPDS)	HepG2 (HLPS)	TPH-1 (LPDS)	BRIN-BD11 (LPDS)
Butyrate	↓ ****	↓ **	↓ **	↑ *
Atorvastatin	↓ ***	↑ *	↓ **	↓ (*p* = 0.17)

**p* < 0.05, ** *p* < 0.01, *** *p* < 0.001, **** *p* < 0.0001.

**Table 2 ijms-23-15506-t002:** Antibodies used in immunoblotting.

Target	Type	Product Code	Company
GAPDH	Mouse monoclonal	ab8245	abcam
HMGCR	Rabbit monoclonal	ab174830	abcam
LDLR	Rabbit polyclonal	ab30532	abcam
SREBP-2	Rabbit polyclonal	ab28482	abcam
APO-A1	Mouse monoclonal	sc376818	Santa Cruz Biotechnology
APO-B	Mouse monoclonal	sc-13538	Santa Cruz Biotechnology
ABCA1	Mouse monoclonal	ab66217	abcam
ABCG1	Rabbit monoclonal	ab52617	abcam
SRB1	Rabbit monoclonal	ab52629	abcam
Caveolin-1	Rabbit polyclonal	ab18199	abcam
Goat Anti-Mouse	IgG H&L (HRP)	ab6789	abcam
Goat Anti-Rabbit	IgG H&L (HRP)	ab6721	abcam

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
