# Peer review of "Butyrate Lowers Cellular Cholesterol through HDAC Inhibition and Impaired SREBP-2 Signalling"

_ijms, 2022, doi:10.3390/ijms232415506_

Round 1

Reviewer 1 Report

The study explores through different experiments the effects of HDAC inhibition on cholesterol metabolism in several cell types (as liver cells, macrophages or insulin-secreting cells), evaluating influence of several drugs on cell cholesterol content, SREBP-2 signalling pathway and cholesterol import/export.

Interestingly describes similarities and differences to statins in some of the actions studied.

The study explores the mechanism of action of sodium butytrate, compared to several statins. It is hypothesized if theses cellular findings could be relevant to the animal or human organisms.

Comments and Questions:

Acronyms and abreviations should be reviewed, and explained the first time it is used in the text (e.g. CVD, SREBP, etc), also avoid labelling in different manners (e.g. atorvastatin – atorv, in the figures. Figures should be self-explaining, thus the footnotes should explain clearly what is represented. Some typos such as repetition of words (furthermore, furthermore, ) should be corrected.

In order to better know about the effects of the drugs on hepatic cells, it would be nice to carry out flow cytometry and confocal microscopy studies to analyse in more detail the LDL receptor expression and cholesterol uptake by HepG2 cells.

Given the importance of HMGCR in the mechanism of action of these drugs, it is not only the gene expression, but preferably the HMGCR activity what would have been measured. Please explain if it was analyzed or not and explain.

Author Response

We thank the reviewer for their comments; there is no doubt it has improved the quality of the paper.

Our responses are listed below in italics.

Comments and Questions:

  1. Acronyms and abreviations should be reviewed, and explained the first time it is used in the text (e.g. CVD, SREBP, etc), also avoid labelling in different manners (e.g. atorvastatin – atorv, in the figures. Figures should be self-explaining, thus the footnotes should explain clearly what is represented. Some typos such as repetition of words (furthermore, furthermore, ) should be corrected.

We apologise for these oversights, which are actually quite embarrassing. We have endeavoured to correct all of these issues in the revised manuscript.

  1. In order to better know about the effects of the drugs on hepatic cells, it would be nice to carry out flow cytometry and confocal microscopy studies to analyse in more detail the LDL receptor expression and cholesterol uptake by HepG2 cells.

Thank you for this suggestion. Given more time, such studies would add further value to the manuscript, but ultimately we do not feel this will add much to the conclusion of the study, and as it relates to the difference between statins and butyrate concerning their effects on cholesterol metabolism, and particular LDL-receptor expression and what this means in terms of their predicted abilities to lower serum LDL-cholesterol concentrations.

  1. Given the importance of HMGCR in the mechanism of action of these drugs, it is not only the gene expression, but preferably the HMGCR activity what would have been measured. Please explain if it was analyzed or not and explain.

Unfortunately given limited resources, this is something which was not done. However, we did measure both HMGCR gene expression, and HMGCR protein expression. Lowering of HMGCR protein expression likely predicts a decrease in activity. It is however something we plan for future studies. 

Finally, we have also shortened the methods section to remove material that we have previously published; i.e, we refer to our previous publications for the details, as suggested by the journal editors to reduce self plagiarism.

Reviewer 2 Report

1. This paper has a lot of abbreviations without appropriate explanations. What CVD stands for? What HMGCR stands for? And what TG stands for? And many others. Please address changes accordingly. 

2. A lot of redundancy is found. On page 5, line 141: remove one "furthermore. On page 10, line 238: remove one "Influence of butyrate on cholesterol-related gene expression in HepG2 cells. "

3. A pathway diagram will be much more intuitive to show how butyrate affects the expression of different proteins. 

4. Do you have any experimental evidence that butyrate alters SREBP-2 activity? How? If no experiment was done (which I think is not a hard experiment to test your hypothesis), do you have any findings from peers' work? 

Author Response

We thank the reviewers for their comments and have amended our paper wherever possible in the time given for our response.

Our responses are listed below in italics.

Reviewer 2

  1. This paper has a lot of abbreviations without appropriate explanations. What CVD stands for? What HMGCR stands for? And what TG stands for? And many others. Please address changes accordingly. 

We apologise for these oversights, which are actually quite embarrassing. We have endeavoured to correct all of these issues in the revised manuscript.

  1. A lot of redundancy is found. On page 5, line 141: remove one "furthermore. On page 10, line 238: remove one "Influence of butyrate on cholesterol-related gene expression in HepG2 cells. "

This has been corrected  in the revised manuscript.

     3.  A pathway diagram will be much more intuitive to show how butyrate affects the expression of different proteins. 

We now include an additional figure (Figure 10) which explains the fundamental difference between the influences of statins and butyrate and the possible consequences for the lowering of serum LDL-cholesterol and which the most pertinent aspect to the possible use of butyrate as a lipid lowering therapeutic.

  1. Do you have any experimental evidence that butyrate alters SREBP-2 activity?

The evidence that we have is indirect, that is, the observation that butyrate treatment reduces the level of the active or cleaved SREBP2 fragment. It is this fragment which binds to the steroid response element (SRE) of numerous cholesterol regulated genes, with the HMGCR and LDLR genes being the best examples in the field. The expression of these genes and their encoded products, which we also measured, is considered to be controlled by SREBP2. The activity of SREBP2 relates to its ability to act as a transcription factor to increase expression of such genes. A decrease in cleaved SREBP2 (i.e. the 68 Kda fragment) can be predicted to result in decreased activity. Studies elsewhere on TSA (another and more potent HDAC inhibitor we also studied) have used similar approaches. There are other studies which can be done, for example of the binding of cleaved/active SREBP2 to the SRE of the HMGCR gene by EMSA (electrophoretic mobility shift assays) and which we now mention but these are quite involved and are for future follow-up studies. The relevant new text from the discussion is shown below (in underlined italics) and in context of the preceding related text.

“The SREBP-2 protein itself can be acetylated and this acetylation alters its localisation and activity [21]. In particular, the class III HDAC SIRT1 deacetylates SREBP-2, and SIRT1 inhibition increases levels of the active nuclear SREBP-2 protein and expression of SREBP-2 target genes [22]. Sodium butyrate does not inhibit class III HDACs [23], and it has not been investigated what effect class I and II HDACs, inhibited by butyrate, have on the acetylation of SREBPs or whether inhibition of class I and II HDACs causes a compensatory increase in class III HDAC activity, and both this and direct evaluation of the binding of active SREBP2 to the steroid response element of HMGCR and LDLR, for example by electrophoretic mobility shift assays are aspects which could be examined in future studies [24].

Finally, we have also shortened the methods section to remove material that we have previously published; i.e, we refer to our previous publications for the details, as suggested by the journal editors to reduce self plagiarism.

Round 2

Reviewer 2 Report

Good to be published.